# The Healthcare Relationship during the Second Wave of the COVID-19 Pandemic: A Qualitative Study in the Emergency Department of an Italian Hospital

**DOI:** 10.3390/ijerph20032072

**Published:** 2023-01-23

**Authors:** Sabrina Cipolletta, Sara Previdi, Simona Martucci

**Affiliations:** Department of General Psychology, University of Padua, 35122 Padova, Italy

**Keywords:** COVID-19 pandemic, doctor–patient relationship, emergency, healthcare system, qualitative research

## Abstract

The COVID-19 pandemic had an impact not only on people’s lives but also on the healthcare system. This study aimed to investigate the healthcare relationship in the Emergency Department (ED) of a hospital in northern Italy, during the second wave of the COVID-19 pandemic. The participants (*N* = 43) consisted of 16 nurses, 6 doctors from the hospital ED, and 21 patients who accessed this department. Semi-structured interviews were carried out and a thematic analysis was conducted. The findings suggest that the COVID-19 pandemic brought both positive and negative changes to the healthcare relationship that are linked to changes in the organization of the ED and to participants’ various experiences of the pandemic. The changes in this relationship should be monitored because they could have long-term effects on healthcare professionals’ wellbeing, treatment outcomes, and the healthcare system. The findings from this study could be used to understand these changes and inform intervention strategies to improve the healthcare relationship.

## 1. Introduction

The healthcare relationship plays a central role in the healthcare process, as it is closely associated with treatment adherence, patient satisfaction, and treatment outcome [1,2]. Trust is a fundamental component of this relationship [3]. High levels of trust are associated with greater cooperation between doctor and patient [4], faster healing rates, and a greater willingness on the part of the patient to accept advice from the doctor [5]. Conversely, low levels of confidence correlate with reduced recovery rates and may lead to patient rejection of diagnosis and treatment [6].

In the specific context of the Emergency Department (ED), the healthcare relationship is affected by several environmental factors that pose significant challenges to doctor–patient communication. Such factors include overcrowding and the continuous reception of patients, a noisy and chaotic environment, physical symptoms complained of by patients, and demands on healthcare professionals that restrict and reduce interactions with patients [7,8,9]. Both the ED and the whole healthcare system were affected by the COVID-19 outbreak. 

The COVID-19 pandemic caused one of the biggest public health crises in the world, which varied in severity between countries [10]. Italy was the first and one of the most affected of the Western countries with more than 17 million cases of the virus, and 166,000 deaths in the first two years of the pandemic [11]. The National Healthcare System in Italy was not prepared to deal with such an emergency. This was revealed by the shortage of individual protection devices, which the World Health Organization (WHO) recommended to reduce the risk of contagion [12], and the lack of hospital beds [13]. During any emergency, in fact, the number of patients who need treatment increases significantly and rapidly, putting a strain on healthcare resources and facilities, as well as professionals [14].

In many European countries, the COVID-19 pandemic drastically reduced patients’ visits to the ED, because they feared infection either in the hospital or on their way to it [13]. It also led to a reorganization of work in EDs, as they were directly involved in the health emergency due to being the first point of access for COVID patients. However, few studies [15,16,17,18] have analyzed the impact of the pandemic on the healthcare relationship. Existing studies have only covered two EDs [19,20] and none in the Italian context. The literature shows that this pandemic altered some aspects of the healthcare relationship, such as non-verbal communication, which was hindered partly by individual safety devices, and physical contact [18]. However, the literature also highlights how patients demonstrated a certain leniency towards healthcare professionals, recognizing how such a stressful period had impacted their job and attributing any oversights or lack of communication to the difficult circumstances rather than to the healthcare professionals themselves [19].

Lavoie et al. [20] underlined the importance of monitoring the effects of working through a protracted pandemic in the long term. The present study has explored these effects for precisely one year after the COVID-19 outbreak and is the first to consider the perspectives of both patients and healthcare professionals in a field study. The aim of this study was to analyze changes in the healthcare relationship during the second wave of the COVID-19 pandemic in an Italian ED, while also taking into account participants’ experiences of the pandemic. 

## 2. Materials and Methods

### 2.1. Participants and Setting

This study was conducted in the ED of a public hospital in northern Italy, where 14 physicians, 53 nurses, and 14 licensed practical nurses worked. The sample included 43 participants: 21 patients (12 women and nine men) aged between 24 and 84 years old (M = 57 years), 6 doctors (three women and 13 men) aged between 32 and 57 years old (M = 45 years), and 16 nurses (seven women and nine men) aged between 28 and 55 years old (M = 40 years). The patients were all from northern Italy and had non-urgent clinical conditions. Patients with painful or high-risk conditions were excluded to avoid disturbing them. The healthcare professionals selected for the interview had all been working in the ED for at least two years. This meant they were able to compare their current experience with that before the pandemic. The doctors had worked in the ED for 12 years on average, and the nurses for 4 years. The sampling took place between April and May 2021 until saturation was reached. This is the point at which the inclusion of any new data does not bring in any more useful information for answering the research questions [21], usually when the number of interviews reaches around 15 (+/−10) [22].

### 2.2. Data Collection

Semi-structured interviews [22] were conducted using two separate interview guides constructed to address the research questions and adapted to the specific needs of patients and healthcare professionals (Table 1) [23]. The conversations with patients explored their experience in the ED, their relationships with nurses and doctors, their personal experiences during the pandemic, and their risk perceptions while accessing the department. The interviews with professionals explored the same areas but from their professional perspectives as well as their work experience during the pandemic. These interviews also explored participants’ views of their professional role and motivations that led them to maintain a certain kind of relationship with patients. Although following a common guide, the interviews were conducted in a flexible and conversational way. The mean interview duration was 27 min for patients (SD = 10.3) and 32 min for healthcare professionals (SD = 7.2). 

A researcher contacted the patients while they were in the ED waiting room to ask them if they were willing to be interviewed once their medical examination was over. Interviews took place in an empty ED room, where it was private and quiet. Another researcher contacted the healthcare professionals at the end of their work shift and, if they consented to participate in the study, an appointment was fixed on the basis of their availability. These interviews took place in the ED meeting room, where they would not be disturbed.

### 2.3. Data Analysis

All interviews were audiotaped and fully transcribed. A thematic analysis was conducted following the six steps proposed by Braun and Clarke [24]: (1) familiarization with the data; (2) generating initial codes; (3) collecting similar codes in overarching themes; (4) reviewing themes; (5) refining and naming themes; (6) constructing a report containing all the citations grouped by code and theme. After reading the interview transcripts several times to obtain a general overview of participants’ experiences and stories, with the aid of ATLAS.ti software, two researchers independently selected and coded the quotations of interest, created code groups (i.e., the themes), identified the central themes, and refined them until the final themes were selected. Coding procedures were continuously discussed with a third researcher, an expert in qualitative studies, to evaluate whether all the themes had sufficient supporting data with the appropriate extracts and to judge whether they met the internal homogeneity and external heterogeneity criteria. Since the authors acknowledged their active role in identifying patterns and themes, reflexivity was carefully sought through repeated comparisons and step-by-step discussions between all the researchers about any possible alternative interpretations of the results. The study was reported according to the Consolidated Criteria for Reporting Qualitative Research [25].

### 2.4. Ethical Considerations

Ethical principles were adhered to throughout the study and all participants signed a written consent form. The study was approved by the Ethics Committee for Psychological Research at the University of Padua and by the head of the ED.

## 3. Results 

Three main themes were identified and they are reported with the corresponding codes in Table 2. The themes were common to the healthcare professionals and patients and eventual differences between them are illustrated, also specifying the participant’s professional role in parentheses after the reported quotation. 

### 3.1. Personal Experiences of the Pandemic

Personal experiences during the pandemic varied among the participants. The majority of health professionals—especially doctors (four out of six)—reported a profound level of stress that impacted on their wellbeing, amplified by the renunciation of holidays and rest periods to ensure staff availability, and the lack of social relations and entertainment these restrictions implied: 


*“All this has led me to no longer considering myself, I don’t know who I am, in the sense that I have given so much to the outside world and I have no longer looked at myself, let’s say, at the priorities that I had. For example, when I am stressed, I eat, even if I am not hungry, in fact I have gained weight compared to before.”*
(P3, F, doctor) 

In addition, female healthcare professionals highlighted how the COVID-19 pandemic upset their family life, causing difficulties in home management and contributing to exhaustion, fatigue, and stress:


*“After work you go home and you have to manage the child’s distance learning and the husband’s smart working, it was heavy.”*
(P14, F, nurse) 

Healthcare professionals also described the difficulties of dealing with COVID-19, it being an unpredictable, unknown, and severe disease that initially they were unsure how to treat. Moreover, the high number of infections and deaths that occurred on a daily basis led some healthcare professionals to experience a failure in their helping role. Six participants (four doctors and two nurses) reported feelings of helplessness, especially in the initial stages of the pandemic; three even questioned their professional choice:


*“I think it was the only time in my life I regretted not becoming a math teacher […] I said, ‘I would be at home doing remote teaching’.”*
(P1, F, doctor) 


*“I felt a bit helpless, because you couldn’t do anything about it; not being able to do anything about it makes you feel a bit helpless, a bit like being without strength because it wasn’t something that, I mean, it failed—the doctor–patient role, because it is not the doctor who masters the situation.”*
(P9, M, doctor) 

On the contrary, seven healthcare professionals experienced the pandemic with serenity, two considered it as an unprecedented situation that could even be “fascinating” (P18, M, nurse), and four as an opportunity to rethink their lives and to understand what is important and meaningful for them:


*“Many things that were taken for granted came to the fore and have been re-appreciated again.”*
(P20, F, nurse) 

The majority of patients highlighted a sense of calmness and wellbeing due to living in a house with a garden or working throughout the lockdown, but also the lack of awareness of what was happening to others. There was a low sense of danger for young people:


*“I am calm because I am young.”*
(P7, M) 


*“I did not think that I could be infected. I hardly thought about that at all!”*
(P9, M) 

The impact of the pandemic on patients’ lives also depended on the coping strategies they used to face it, such as accepting the situation and accepting their fate, following the rules and restrictions, trying to live without fear, detaching themselves from social media and the television news about COVID-19, and looking at the positive side of the pandemic. Some patients also described the virus as an opportunity to make time for themselves, to find peace, learn to understand people from different perspectives, and evaluate which personal relationships mattered most to them:


*“The limitations have re-humanized some relationships and downsized others.”*
(P20, M) 


*“If you put the fear on you, you do not move anymore out of your home. Thereby, I tried to live serenely because we already have many problems in our lives so that, it is ok to be careful, but…when some friends came to meet us, wearing masks, windows open, keeping distances but we were not closed, if someone came, we allowed them in.”*
(P17, F) 

Patients and healthcare professionals reported feelings of anger and disappointment with the attitudes of the population in the second phase of the pandemic, such as non-compliance with the rules, negation, and the stigmatization of healthcare professionals as virus-smearers:


*“There is indifference. I mean that it is the public, the people who come to the ED, that make me worry.”*
(P1, M) 


*“More than anything else there is no respect of those who are working, of the rules.”*
(P19, F) 

Six patients also expressed their disappointment with the way the emergency was managed by the government: 


*“Yes, but they are managers, they should be ready for these emergencies, if there is a virus, you have to take action! (…) To be honest, there was a bit of a light management and now we are seeing the consequences.”*
(P14, F) 

Nine healthcare professionals reported a shift in attitudes towards them. Whereas in the first phase of the pandemic they were considered “heroes”, in the second phase, mistrust and disinvestment that was usual before the pandemic came back and healthcare professionals were also identified as “infectors”, when hospitals were perceived as the main source of infection.


*“It bothered me during the first wave, the glorification of the health system, because it is cloying, it seems that you are God, it was only a month before when everyone threw shit at you. Even the political class made cut after cut to the National Healthcare System, then instead we became heroes. I have experienced this thing here quite badly, it bothered me so much, just a feeling of anger.”*
(P9, M, doctor) 


*“There has been some bullshit because if there is contamination, this comes from the hospitals and there is no control. If you know that there is, you take some precautions! There has been a certain imprudence, an imprudence that has been fully charged to the hospitals.”*
(P16, F) 


*“They knew I worked in the hospital, at that time it was difficult to even find people who were willing to come home, because you know you look like an infector, don’t you? Because you work in the hospital.”*
(P1, F, doctor) 

### 3.2. Pandemic Experience in the ED

All healthcare professionals described the pandemic experience in the department as demanding due to the increased workload, initial unpreparedness, and changes in the department’s organization. They complained about the initial absence of guidelines and protocols, the reorganization of the department into COVID and non-COVID areas, the new staff hired to work in the COVID area (three months after the beginning of the emergency), and new protocols that generated confusion: 


*“It was very hard work at the outset and also in the second wave, it was a very heavy workload, enormous in fact, and from an organizational and logistical point of view it was sometimes chaotic as well as heavy.”*
(P12, M, nurse) 

On the contrary, patients perceived the re-organization of the ED positively, underlining a general improvement in the ED’s management and shorter waiting times compared to previously:


*“In my opinion there is much more organization now.”*
(P21, M) 

After an initial shortage of personal protective equipment (PPE), healthcare professionals claimed to have all the necessary means available to protect themselves from the virus and stated that the use of protective devices had fostered a feeling of safety during the performance of their work, despite the fact that using them caused other difficulties:


*“I was frustrated because it was heavy, I always felt harnessed, as if in a cage”*
(P4, M, nurse) 

Wearing masks was also a cause of discomfort for some patients during visits to the emergency room: 


*“You can’t tell a person who is out of breath ‘put a mask on!’ You will kill them! I was dying, I wasn’t breathing anymore! (…) Yes, prioritize urgency first! COVID is urgent but, at this point, I am dying! Why should I care about COVID?”*
(P2, F) 

Protective measures also brought a sense of safety and calm to some patients. However, fear of accessing the department and of spending time in the waiting room was something that was reported often. Patients trusted the healthcare professionals but feared contact with other patients. This fear led one patient to delay access to the department:


*“I’ve been sick since Monday (…) I immediately recognized the symptoms; I went to the general practitioner, and he said, ‘go to the emergency room’ and I said ‘no’.”*
(P13, F) 

Some patients developed strategies to reduce their contact with other patients as much as possible, such as by isolating themselves in a corner of the waiting room or by spending their waiting time in the hospital cafe instead of staying in the designated area. Other patients, especially those who were chronically ill, stressed the fact that acute pain and high risk of complications forced them to access the emergency room and prioritize their own health over the fear of contracting the virus.

More than half of the healthcare professionals also reported being afraid of contracting the virus at work, especially in the initial phase of the health emergency. Three said they were fearful when working in an ambulance, it being a small and tight environment, and when going to unknown places with equally unknown people where the risk of coming into contact with COVID-positive people was greater, especially because some patients did not report having the virus. One doctor said:


*“I was afraid of getting sick, of dying. I had never thought about dying from something I could have got here.”*
(P2, F, doctor) 

All healthcare professionals reported that over time they became accustomed to cohabiting with the presence of the virus in the department. Five healthcare professionals said that adapting to this critical situation was probably easier for them as they were used to coping with emergency situations and the risk of infection. They even chose to work in the COVID unit due to its dynamism and in order to test themselves:


*“Later, having people infected [with COVID] in here became integrated into the normal routine and, therefore, the tension levels dropped (…) then we became used to having to deal with critical situations and this meant that adaptation times were speeded up.”*
(P13, F, nurse) 


*“When a person decides to do our job, they are already aware of what they might face, today COVID, tomorrow another disease, or a disease due to a virus, such as HIV.”*
(P10, M, nurse) 

### 3.3. Healthcare Relationships during the Pandemic

Many participants mentioned limitations on the healthcare relationship caused by using protective devices. About half of the healthcare professionals said that the use of some devices represented a barrier, making human contact, relationships, and communication more challenging, especially with older people:


*“Shielding was not only a barrier for the virus. It was also a barrier for human contact. Some contact ceased and, for some, contact with family members ceased, which is fundamental but is now missing. Patient treatment became completely dehumanized and was one of the most dramatic changes compared to how we usually did it.”*
(P21, M, nurse) 

In addition, healthcare professionals stressed that the use of aids resulted in a loss of information from non-verbal channels, leading to misunderstandings. Patients also underlined how the difficulty in recognizing doctors’ and nurses’ faces added to the acoustic difficulties caused by masks, which brought new challenges to the relationship:


*“With the mask, words are lost, the speech is lost a little, so you always have to ask ‘how’ and ‘what’ and this bothers me a little, also I am somewhat diminished in hearing, and age makes you lose something, too. So, you have to ask, ‘what did you say, what?’”*
(P10, M) 

Eleven healthcare professionals reported feeling a physical or emotional detachment towards the patients:


*“The relationship with the patient was alienating.”*
(P2, F, doctor) 

This was due to communication difficulties, fatigue, and the need to maintain social distancing to comply with the regulations, but also to avoid the risk of infecting the patient or contracting the virus from them, because all patients who entered the ED were treated as a potential source of infection. Two healthcare professionals said they felt safer approaching patients once they and the patients had received the vaccine. 

One patient highlighted fear in revealing his symptoms, because he thought they would be mistaken for COVID-19 symptoms, which caused him anxiety during the triage process and hesitancy in disclosing his clinical information to the nurse. Six patients highlighted the lack of attention from doctors and nurses expressed by the fact that some healthcare professionals chatted among themselves during their visit, did not make eye contact with them, and showed them little respect. This perception is in contrast to patients’ expectations when accessing the emergency room: 


*“When they put me on the bed during the visit, I didn’t want to be spoiled, however, they showed little respect for the person on the bed. I did not like it. It was very cold. They chatted among themselves, without any thought for me, without looking at me or looking me in the face. Maybe that’s normal there (…). In those moments you see that one patient is the same as any other, they are all the same (…) but there should be more closeness shown to the patient because whoever comes here is sick.”*
(P18, F) 

Another issue often raised by patients was the poor communication and lack of information. Some patients expected the doctors to be more thorough in their explanations regarding their diagnosis and about the results of their examinations. They underlined a sense of confusion and impotence about their clinical situation, while also stressing the desire to be more reassured by healthcare professionals:


*“Well, maybe doctors don’t talk that much, it’s probably normal for them, yes, maybe they should talk a little more, I asked questions and he replied (…) but if a patient does not ask questions the doctor should still explain or speak anyway (…) it all seems obvious right? No, it is not obvious. Especially for someone who is in the ED, it would be nice to feel reassured for a moment.”*
(P19, F) 

Contrary to the experiences previously mentioned, two patients said they noticed improvements in the healthcare relationship, such as greater attention given to them, along with more humanity and empathy: 


*“I remember from years ago that not only doctors, but also grumpy employees, those who said ‘Yes, well, come here, I have to do it, so I’ll do it’ (…) they treated you a bit like veterinarians treat certain animals. Now (…) you feel that they are human!”*
(P20, M) 

Three healthcare professionals also believed that due to the pandemic, the relationship with the patient had improved. They reported better communication, a more direct relationship with the patient, and slightly more time dedicated to them than previously.


*“Only the patient can enter, so the caregiver is left outside, maybe the relationship is a little more direct because maybe there isn’t the obstacle of the relative who wants to add something or describe the situation differently.”*
(P15, M, nurse) 

Patients often understood and justified healthcare professionals’ stress and fatigue due to the challenges that the pandemic brought to their jobs and the risks that they faced every day. Seven healthcare professionals said there were fewer aggressive and arrogant patients and that there was more respect shown to healthcare personnel, but five of them said this only changed in the initial phase of the pandemic and was associated with the peak of the emergency.


*“Patients have begun to understand, they thank us more, they are less arrogant, but this unfortunately depends on the [pandemic] waves: during the peaks of the pandemic the patients who are really sick come and say, ‘I’m sorry, I came, however, because I was really sick!’ The arrogant people who complain when you explain to them that their problem is not an urgent one were much less.”*
(P11, M, doctor) 

In total, 20 of the 22 healthcare professionals reported satisfaction in their relationships with users, and defined themselves as capable of communicating effectively with patients to find a solution to their problems. Additionally, 19 healthcare professionals reported feeling satisfied with their relationships with patients because they received positive feedback from the patients themselves. These participants considered themselves attentive to their relationships with patients because they believed it to be an integral part of the care process, allowing for greater collaboration, better patient compliance, and greater personal satisfaction:


*“The comparison with the person in front of me is most important for me. Yes, there is the disease, but first of all there is the person.”*
(P10, M, nurse) 


*“Honestly, it is not that this pandemic has changed the way I see the patient and interface with him, so how I behaved before, I behave now. Honestly, the patient is the same before and after COVID, and my relationship with him is the same too.”*
(P5, M, doctor) 

Only two healthcare professionals were not satisfied with their relationships with patients during the pandemic due to time constraints and the high number of patients requiring treatment. They were the same two who reported being more focused on the patients’ disease than on their relationship with them:


*“I don’t even consider that relational part of taking charge (…) unfortunately, the way I work in the ED is in an emergency situation, and therefore, the relationship fails.”*
(P7, M, nurse) 

Generally, the feeling of being listened to by healthcare professionals was essential to patients’ satisfaction. When healthcare professionals asked precise questions, and actively tried to understand patients’ clinical histories or their conditions and gave advice, patients felt cared for:


*“[The doctor] has been very kind and I noticed that she also worried about me, she took care to tell me ‘Look, you could do [it] in this way’. Do you understand what I mean? That’s ok in my opinion, I do not pretend who knows what, but a word!”*
(P18, F) 

## 4. Discussion

This study explored the healthcare relationship in the ED of a hospital in northern Italy during the second wave of the COVID-19 pandemic. The findings have shown that the pandemic produced changes in the healthcare relationship that are linked to the changes in the organization of the ED and to participants’ various experiences of the COVID-19 pandemic. These links are illustrated in Figure 1 and discussed below.

Individuals and communities responded in various ways to the COVID-19 pandemic with different psychological and social consequences [26,27]. The literature shows how acceptance, avoidance of negative thoughts, living without fear, meaning making, and optimism proved to be useful for maintaining wellbeing and acted as a protective factor for psychological and psychiatric disorders [28,29]. The participants—both healthcare professionals and patients—who reported these strategies and had positive experiences of the pandemic also reported a positive experience in the ED and showed no particular fear of becoming infected [28]. Conversely, patients who reported experiencing anxiety and stress in their personal lives experienced fear of infection while in the waiting room and were more likely to delay access to the hospital, which has been a common phenomenon during the pandemic [30,31]. Likewise, the healthcare professionals (the majority in our sample) who reported a detriment in their wellbeing and social relationships, a sense of helplessness, and even doubts about their professional choice also expressed more difficulty in facing the pandemic situation in the ED. These participants were mainly doctors, who were also older than nurses and had been working in the ED for longer. All these aspects, together with the overload of responsibilities concentrated on a single doctor, might have contributed to their suffering.

Other aspects of participants’ experiences of the pandemic that affected the healthcare relationship were the omission of information in communications on the part of patients due to the fear of being considered a vehicle of the virus that could compromise the accuracy of diagnosis and choice of treatment [32] and the fatigue felt by all healthcare professionals, which implied less empathy towards patients already existed in the pre-pandemic period [33,34,35]. Physical and emotional exhaustion are inversely proportional to empathy [36]; therefore, it is not surprising that those healthcare professionals who reported emotional detachment towards the patients were also those who complained of greater levels of fatigue.

Both healthcare professionals and patients also pointed out the change in people’s attitudes towards healthcare professionals one year after the COVID-19 outbreak that implied changes in the healthcare relationship. As already pointed out in the literature, the professionalism shown by healthcare professionals in the face of the health emergency [37] and the predominant narrative of doctors and nurses as “heroes” in the media fostered empathy and respect towards them [26,38] and the National Healthcare System in the first wave of the pandemic [39]. In addition, patients were less aggressive and displayed fewer violent attitudes, which improved doctor–patient communication as well as greater compliance with care. However, during the second wave of the pandemic, mistrust and disinvestment towards healthcare professionals, which was common before the pandemic, returned and some people assumed an attitude of neglect in their inadequate use of safety devices, their non-compliance with quarantines, or by accessing the emergency room for non-urgent problems. In line with the results of the studies conducted during the first wave of the pandemic [40,41,42], this attitude was reported by the participants in the present study and caused anger and disappointment from health professionals, because it was interpreted by them as if people no longer recognized the efforts they were making during the emergency. Additionally, the participants reported that some people began to define healthcare professionals as infectors rather than heroes, with relevant implications in these professionals’ personal lives and in the healthcare relationship. These negative attitudes undermine the trust that is a fundamental component of this relationship [3] with negative consequences for treatment adherence and doctor–patient collaboration [4,6]. On the contrary, most of the patients involved in this study expressed gratitude towards the healthcare professionals and understood and justified some gaps in their communication as being caused by healthcare professionals’ stress and fatigue due to the challenges that the pandemic brought to them.

Some of the changes in the ED introduced by the pandemic also impacted the healthcare relationship. The use of protective devices and physical distancing recommended to contain the contagion made participants feel safer but were also experienced as a limitation in the healthcare relationship. In line with the results of the studies conducted during the first wave of the pandemic [18,19,32,43,44,45], participants reported that the use of personal protective devices prevented precise recognition of the person in front of them, limited facial expressions, and altered verbal as well as non-verbal communication. Not being able to see the face of the professionals negatively affected empathy and perceived attention. Moreover, masks caused a significant degradation of the acoustic signal, which, if added to high noise levels present in the hospital setting, can generate psychological distress for patients. Physical distancing increased the relational distance between healthcare professionals and patients and some participants also stated, due to the fear of contagion but also to provide greater protection to the patient and adhere strictly to the anti-contagion rules, that they avoided, when possible, any physical contact at all. As a consequence of this physical distancing, reassurance becomes more difficult, because, as reported by some healthcare professionals, physical gestures that convey safety and confidence, such as a hand on the shoulder or a handshake, are no longer possible. 

Some patients, however, noticed an improvement in the healthcare relationship during the pandemic in terms of more attention given to the patient and humanity during visits. Greater attention given to the patient might have been a strategy used by healthcare professionals to overcome the barriers imposed by the preventive measures, as suggested by Sugg et al. [46]. This, alongside less crowding in the emergency room, due to the reduction in visits and access to wards for fear of spreading the virus, and the hiring of new healthcare professionals to strengthen the human resources of the National Health System, fostered, in some phases of the pandemic, better care of patients. This included slightly longer doctor–patient interactions and shorter waiting times [47]. Some healthcare professionals also highlighted how the absence of caregivers and their interruptions and requests during medical examinations allowed them to have a more direct relationship with those patients who were self-sufficient and capable of understanding them [48,49]. 

Finally, the changes observed by healthcare professionals about their relationships with patients during the pandemic were influenced by the role they attributed to their relationships with users in the care process [50,51]. Those healthcare professionals who defined themselves as problem-oriented rather than patient-oriented during their working practice also maintained greater distance in the relationship during the pandemic. On the contrary, those who claimed to be attentive to communication and their relationships with patients complained about the limitations brought to the relationship by the changes due to the COVID-19 pandemic and tried to maintain humanity and good communication with the patients. No differences were found in this aspect between doctors and nurses, whereas a gender difference was noticed in the fact that women were more sensitive and attentive to the relationship, as previous literature has already pointed out [52].

The main limitation of the present study is the specificity of the sample: all participants came from the same geographical area and were attending the same ED. This is in keeping with the aims of qualitative research to explore people’s experiences within their context rather than the provision of findings that can be generalized [53]. Nevertheless, future studies might explore a range of experiences with different people in a variety of contexts. Another limitation is due to the sample consisting of a greater number of nurses than doctors, which was reflective of the proportion in the ward, but has underrepresented the doctors’ perspectives. Although the only difference that was found between their perspectives was in the level of suffering, further studies might further explore other eventual differences.

## 5. Conclusions

The results of this study shed light on the impact that changes introduced due to the COVID-19 pandemic may have in the long-term on the wellbeing of healthcare professionals, on treatment outcomes, and on the healthcare system. This is the first study to explore the impact of these changes from the perspectives of professionals and patients who experienced them first hand in an ED. Their accounts of their experiences were collected on-site immediately after the changes took place, thus providing a more accurate recollection.

The findings suggest that the COVID-19 pandemic brought about both positive and negative changes to the healthcare relationship in the ED. However, the findings also show how healthcare professionals in the emergency room of the hospital are confronted daily with various occupational risks and critical working situations due to the peculiar environmental factors when compared to other departments. In the long term, this can lead to professionals becoming physically and emotionally exhausted. Therefore, it would be advisable to promptly promote psychological intervention to improve their wellbeing and the quality of care.

As pointed out in a recent review [54], these interventions should not only be targeted to prevent or relieve mental health problems at an individual level, but must also act at an organizational level to allow healthcare professionals to feel safe and supported. The results of the present study showed some protective factors, such as access to adequate personal protective device, transparency in communication, and trust and availability in the relationship. At the same time, some barriers, such as work overload, detachment in the relationship, distress, and mistrust, may hinder healthcare professionals’ wellbeing and the quality of the relationship with patients. All these aspects should be taken into consideration to inform policy makers in possible strategic directional choices. Finally, the changes in the healthcare relationship should be monitored, because taking care of the relationship is one way to take care of the patient, the caregiver, and the healthcare context.

## Figures and Tables

**Figure 1 ijerph-20-02072-f001:**
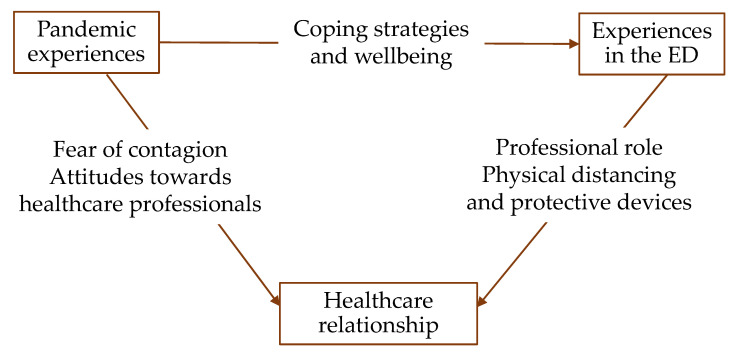
Map of the relationships between themes.

**Table 1 ijerph-20-02072-t001:** Interview guides.

Interview Guide with Patients
Can you describe your experience in the emergency department? How would you describe your relationship with the doctors and nurses who visited you? Has the pandemic influenced your decision to access the emergency department? How? What is your experience of the pandemic in your daily life? If you have been to the emergency department at other times before the start of the pandemic, how did you experience it then?
Interview Guide with Healthcare Professionals
Can you describe your experience in the emergency department? How would you describe your relationship with users? How does the current pandemic affect your working conditions? And your personal life? If you compare the relationship you had with the patients prior to the onset of the pandemic with the relationship you have with them today, during the pandemic, do you think there have been any changes? In what aspects?

**Table 2 ijerph-20-02072-t002:** Themes and codes for the narratives of healthcare professionals and patients.

Themes	Codes	
	Patients	Healthcare Professionals
Personal experiences of the pandemic	Importance of housing situation Confusion and lack of information Belief in conspiracy theories Anger and disappointment about the attitudes of the population Disappointment in the government’s management of the pandemic Caregiving during the pandemic Calmness Coping strategies Vaccine experience	COVID-19 as an unknown disease Feelings of helplessness Disappointment and anger at the attitudes of the population Fear of contagion Doubting your work COVID-19 as an unprecedented situation COVID-19 as an opportunity of change Personal well-being Vaccine as a weapon to protect oneself and others
The pandemic experience in the emergency department	Re-organization of the ED Shorter waiting times Discomfort due to protective masks Sense of safety due to protection devices and social distancing Fear of infection in the waiting room and in the department Delayed access	Tiring and demanding experience Personal protective equipment obligation Habits and coexistence with the virus
The healthcare relationship during the pandemic	Limitations in relationships due to protective devices Fear in revealing symptoms Improvement in relationships Accounting for healthcare professionals’ stress and fatigue Justifying healthcare professionals Importance of feeling listened to Lack of attention Lack of clear information Need to be reassured	Safety devices as an obstacle to relationships Emotional and physical detachment from patients Relationship improvement No changes Risk as an integral part of the job Motivation for working in an emergency department Working satisfaction Role attributed to relationships with patients

## Data Availability

Due to the nature of this research, participants of this study did not agree for their data to be shared publicly, so supporting data is not available.

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
