# Peer review of "The Healthcare Relationship during the Second Wave of the COVID-19 Pandemic: A Qualitative Study in the Emergency Department of an Italian Hospital"

_ijerph, 2023, doi:10.3390/ijerph20032072_

Round 1
Reviewer 1 Report
The sample is unrepresentative. The outcome does not provide great tools for policy mackers in possible strategic directional choices.
Author Response
Thank you for reviewing the manuscript. We understand that from a quantitative perspective the sample is unrepresentative but from a qualitative perspective it is sufficient to satisfy the criteria of theoretical saturation, which is coherent with a qualitative approach, as specified on p.3 line 76-77. The aim of this sampling method is not to be representative of a population but to allow to understand a specific context. Forty participants are in the usual range of a qualitative study if you consider that usually the number of interviews is around 15 (+/-10) (Kvale, 1996; Reid, Flowers, & Larkin, 2005). We have now specified this in the manuscript.
Pag. 2-3 line 77-78: The sampling took place between April and May 2021 until saturation was reached. This is the point at which the inclusion of any new data does not bring in any more useful information for answering the research questions [21], usually when the number of interviews reaches around 15 (+/−10) [22].
The results offer indications for policy makers in possible strategic directional choices as they illustrate personal and social criticalities experienced by healthcare professionals and patients during the pandemic. We have now made this more explicit in the conclusion.
p.13-14 line460-467: As pointed out in a recent review [54] these interventions should not only be targeted to prevent or relieve mental health problems at an individual level, but must also act at an organizational level to allow healthcare professionals to feel safe and supported. The results of the present study showed some protective factors such as access to adequate personal protective device, transparency in the communication, trust and availability in the relationship. At the same time some barriers such as work overload, detachment in the relationship, distress and mistrust may hinder healthcare professionals’ wellbeing and the quality of the relationship with patients. All these aspects should be taken into consideration to inform policy makers in possible strategic directional choices.
Reviewer 2 Report
- Overall a good piece of work.
- Explain briefly how ATLAS.ti help in conducting thematic analysis. Also, it will be useful to explain what thematic analysis is in a sentence or two.
- Remove bullet points from the table and some grammatical mistakes to be fixed.
Author Response
Thank you for reviewing the manuscript. We are happy that you appreciated the study reported in the manuscript. Nevertheless, you raised some questions that need to be addressed. We have made the revisions required and responded to your comments and suggestions. In the uploaded file the revisions are in track change modality. We also list all the changes in this letter. Please find below point-by-point responses to the comments. In our response we underline the page of the manuscript corresponding to the modifications performed.
We have explained how ATLAS.ti helped in conducting thematic analysis and explained what thematic analysis is.
p.4: A thematic analysis was conducted following the six steps proposed by Braun and Clarke [24]: (1) familiarization with the data; (2) generating initial codes; (3) collecting similar codes in overarching themes; (4) reviewing themes; (5) refining and naming themes; (6) constructing a report containing all the citations grouped by code, sub-theme and theme. After reading the interview transcripts several times to obtain a general overview of participants’ experiences and stories, with the aid of ATLAS.ti software two researchers independently selected and coded the quotations of interest, created code groups (i.e. the themes), identified the central themes and refined them until the final themes were selected.
We have removed bullet points from the table and fixed the grammatical mistakes.